# Effect of 3-Hydroxyvalerate Content on Thermal, Mechanical, and Rheological Properties of Poly(3-hydroxybutyrate-co-3-hydroxyvalerate) Biopolymers Produced from Fermented Dairy Manure

**DOI:** 10.3390/polym14194140

**Published:** 2022-10-03

**Authors:** Maryam Abbasi, Dikshya Pokhrel, Erik R. Coats, Nicholas M. Guho, Armando G. McDonald

**Affiliations:** 1Department of Forest, Rangeland and Fire Sciences, University of Idaho, Moscow, ID 83844-1133, USA; 2Department of Civil and Environmental Engineering, University of Idaho, Moscow, ID 83844-1022, USA

**Keywords:** poly(3-hydroxybutyrate-co-3-hydroxyvalerate) (PHBV), mixed microbial consortia (MMC), CHCl_3_ extraction, comonomer composition, thermal properties, rheology, mechanical properties

## Abstract

Poly(3-hydroxybutyrate-co-3-hydroxyvalerate) (PHBV) with various 3-hydroxyvalerate (3HV) contents biosynthesized by mixed microbial consortia (MMC) fed fermented dairy manure at the large-scale level was assessed over a 3-month period. The thermal, mechanical, and rheological behavior and the chemical structure of the extracted PHBV biopolymers were studied. The recovery of crude PHBV extracted in a large Soxhlet extractor with CHCl_3_ for 24 h ranged between 20.6% to 31.8% and purified to yield between 8.9% to 26.9% all based on original biomass. ^13^C-NMR spectroscopy revealed that the extracted PHBVs have a random distribution of 3HV and 3-hydroxybutyrate (3HB) units and with 3HV content between 16% and 24%. The glass transition temperature (*T_g_*) of the extracted PHBVs varied between −0.7 and −7.4 °C. Some of the extracted PHBVs showed two melting temperatures (*T_m_*) which the lower *T_m_*_1_ ranged between 126.1 °C and 159.7 °C and the higher *T_m_*_2_ varied between 152.1 °C and 170.1 °C. The weight average molar mass of extracted PHBVs was wide ranging from 6.49 × 10^5^ g·mol^−1^ to 28.0 × 10^5^ g·mol^−1^. The flexural and tensile properties were also determined. The extracted polymers showed a reverse relationship between the 3HV content and Young’s modulus, tensile strength, flexural modulus, and flexural strength properties.

## 1. Introduction

Polyhydroxyalkanoates (PHA) are natural polyesters of hydroxyalkanoate (HA) building blocks that are produced and accumulated intracellularly by numerous bacteria as an energy reserve material in the presence of excess carbon source and/or under limited nutritional conditions [1,2,3,4,5]. PHAs are biodegradable bioplastics and have mechanical properties similar to those of conventional thermoplastics, such as polyethylene (PE) and polypropylene (PP), and are alternates for petroleum-derived polymers [1,6,7]. Poly-3-hydroxybutyrate (PHB) was the first known PHA and as a consequence the most extensively studied PHA [6,7]. Some inherent limitations of PHB including high brittleness and low thermal stability have hampered its commercialization [7,8,9,10]. Adding plasticizers, and low molecular weight additives is an effective way to improve the flexibility and processing window of PHB [10]. Plasticizers penetrate into the polymer network to reduce deformation tension, elastic modulus, and viscosity of a polymer to increase polymer chain flexibility, resistance to fracture, and dielectric constant [10].

Improving the flexibility of PHB can be achieved by the introduction of HA comonomers, such as 3-hydroxyvalerate (3HV), to form the copolymer poly-3-hydroxybutyrate-co-3-hydroxyvalerate (PHBV). Compared to PHB, PHBV is less crystalline and as a result is more flexible with improved impact resistance and toughness which makes PHBV suitable for applications such as flexible packaging [6,10]. The performance of PHBV highly depends on its 3HV content. Higher 3HV content reduces the polymer’s crystallinity, increases its flexibility, and results in a more easily biodegraded polymer [8,10]. An increase in the 3HV content of the PHBV from 0 to 50% can also significantly reduce its melting point (*T_m_*) [7].

Current commercial production of PHBV uses pure bacterial culture and/or genetically modified strains in a sterile environment. Furthermore, commercial production uses refined carbon substrates such as glucose and fatty acids which also increase production costs [3,8,11]. Recently, PHBV production processes using mixed microbial cultures (MMCs) have attracted more attention due to having a simplified operations and using non-sterile conditions allowing for a potentially sustainable and closed-cycle process for polymer synthesis [6,11]. Using a renewable waste carbon substrate and MMCs for PHBV biosynthesis can reduce the production expenses [3,12,13,14]. Depending on the type of renewable waste carbon used as substrate, the type of bacteria and the applied production parameters the production expenses can be reduced by greater than 50% [3,12,13,14,15]. The carbon feed substrate can influence the structure and properties of PHBV [6,16]. PHBV can be synthesized using fatty acids with even (e.g., acetate and butyrate) and odd (e.g., propionate and valerate) numbered carbon chains [6]. The even numbered fatty acids will produce 3-hydroxy butyrate (3HB) while the odd numbered fatty acids will produce 3HV. Therefore, the 3HV to 3HB ratio in PHBV will be dependent on the carbon feedstock [10]. Studies have focused PHBV production parameters and optimizing the extraction and recovery yield in small scale (≤3 g) [17,18,19,20,21] or characterizing the properties of PHBV [22,23,24]. However, there is a gap in comprehensive studies examining PHBV biosynthesis using MMC and various feed compositions, extraction of PHBV, and polymer properties at the pilot scale. 

This study aims were to (i) produce PHBV with different 3HV contents using MMC that are fed with fermented dairy manure liquor over a several month operating period [6], (ii) determine PHBV chemical composition, (iii) determine their mechanical properties, and (iv) determine their thermal and rheological properties. These results show how PHBV properties will vary in a pilot plant over a 3-month operation. 

## 2. Materials and Methods

### 2.1. Materials 

Chloroform (CHCl_3_, 99.8%) was obtained from EMD Millipore, and acetone, petroleum ether, and methanol were technical grade. Two commercial PHBV samples (PHBV-8%HV and PHBV-12%HV (Tianan Biologic Materials C. Ltd., Ningbo, China)) were used as reference materials. PHBV rich biomass samples were produced in a pilot-scale bioreactor over eight production periods (operational day 15, 32, 39, 46, 61,71, 78, and 84) consisting of (i) sequencing batch dairy manure fermenter (760 L) to produce volatile fatty acids (VFA), (ii) an enrichment reactor (680 L) which was fed VFAs to produce an MMC culture that can produce PHBV via a feast-famine regime and (iii) PHBV production reactor (450 L) which was fed VFAs to produce and accumulate PHBV [6]. The 3HV molar ratio and the total amount of PHBV in microbial biomass are listed in Table 1.

### 2.2. Methods 

#### 2.2.1. PHBV Extraction

Purified PHBV was recovered from the PHBV-rich biomass produced at the PHBV pilot facility following the extraction and purification. Frozen PHBV-rich biomass (1–2 kg) of each operational day was lyophilized. To remove lipids, the lyophilized biomass (165–260 g) was acetone washed in three steps (24 h, 5 L) stirring continuously, recovered by vacuum filtration, and dried in air. Lipid-free PHBV-rich biomass was Soxhlet extracted for 24 h with CHCl_3_ (3.5 L) then filtered under vacuum to remove any residual biomass and the extract was concentrated to dryness (rotary evaporator) to obtain crude PHBV. Each of the crude PHBV samples were dissolved in CHCl_3_ (100 mL) and added dropwise to cold (4 °C) petroleum ether (1.5 L) with constant stirring to precipitate the PHBV, recovered by filtration and dried. The recovered PHBV was further purified by dissolving in CHCl_3_ (100 mL) and added dropwise to cold (4 °C) methanol (1.5 L) with constant stirring to precipitate the purified PHBV, recovered by filtration, dried under vacuum, and yield recorded. The crude and purified PHBV yields are given in Table 1. The sample code name for purified PHBV obtained at an operational day will coded PHBV-xx (e.g., PHBV-15—PHBV at 15th operational day).

#### 2.2.2. Gas Chromatography-Mass Spectrometry (GC-MS)

Lyophilized biomass samples (20 mg) were methanolyzed in 3% *v/v* H_2_SO_4_ in methanol (2 mL) plus CHCl_3_ (2 mL, containing 0.25 mg/mL benzoic acid as internal standard) at 100 °C for 4 h. After cooling, water was added (2 mL), the solution mixed, and the CHCl_3_ layer separated and transferred to a vial. The resulting methyl ester derivatives were analyzed by GC-MS (ISQ7000-Trace1300, Thermo-Scientific, Madison, WI, USA) [6]. Separation was achieved using a ZB-1 capillary column (Phenomenex, 30 m, 0.25 mm) and a temperature profile of 40 °C (1 min) ramped to 200 °C (10 min) at 5 °C/min. Data was collected and analyzed using Xcalibur v4.1.31.9 software (Thermo-Scientific, Madison, WI, USA).

#### 2.2.3. Size Exclusion Chromatography (SEC) 

Number average molar mass (*M_n_*), weight average molar mass (*M_w_*), and polydispersity index (*M_w_/M_n_*) of the PHBV samples (2 mg/mL) were determined by size exclusion chromatography (SEC) using a Jordi DVB linear mixed bed column (7.8 mm × 300 mm^2^) on elution with CHCl_3_ (0.5 mL/min) equipped with a Waters model 2414 refractive index detector and a PostNova PN3609 (658 nm) multi-angle light scattering (MALS) detector (Postnova Analytics GmbH, Landsberg, Germany) at 30 °C. The system was calibrated using a narrow polystyrene standard (Malvern, *M_w_* = 105,268·g/mol). Data were analyzed using the NovaSEC v1.5.0 software (Postnova Analytics GmbH, Landsberg, Germany).

#### 2.2.4. FTIR Spectroscopy

Fourier-Transform Infrared Spectroscopy (FTIR) was performed on PHBV samples, in duplicate, using a Nicolet-iS5 spectrometer (Thermo-Scientific, Madison, WI, USA) equipped with an attenuated total reflectance (ATR) accessory (ZnSe, ID5). Spectra were baseline corrected, averaged, and analyzed using Omnic v9.5 software (Thermo-Scientific, Madison, WI, USA). The carbonyl index (*I_C=O_ A*_1720_*/A*_1740_) was determined on normalized spectra that were curve-fitted using Igor Pro 9 software (WaveMetrics, Portland, OR, USA) [25,26]. The area (*A*) of each carbonyl band at 1720 cm^−1^ and 1740 cm^−1^ were Gaussian curve fitted using a peak width at half height of 10 cm^−1^ and integrated. 

#### 2.2.5. Thermal Analysis

Differential scanning calorimetry (DSC) was performed on purified PHBV (4–6 mg, in duplicate using Tzero^TM^ aluminum pans) using a TA instrument model Q200 DSC (New Castle, DE, USA) with refrigerated cooling under nitrogen (50 mL/min). The DSC was calibrated using indium and sapphire standards using following manufacturer protocols. Samples were equilibrated at 40 °C (3 min) then ramped at 10 °C/min to 180 °C (3 min) to remove any thermal history, then cooled to −50 °C at −10 °C/min (3 min), and the cycle repeated. The glass transition temperature (*T_g_*), melting temperature (*T_m_*), and the melting enthalpy (∆*H_f_*) were determined from the inflection point and endothermal peak of the second heating scan, respectively. The degree of crystallinity (*X_c_*) was determined using the following Equation (1)
(1)$$X_{C}\;=\;\frac{\Delta H_{f}}{\Delta H_{f0}}\;\times \;100$$
where ∆*H_f_* indicates the melting enthalpy of the sample, and ∆*H^0^_f_* shows the melting enthalpy of the pure crystalline polymer (146 J/g for PHBV) [27]. Data were analyzed using a TA Universal analysis v4.5A software. The thermal stability of purified PHBV was determined by thermogravimetric analysis (TGA) using a Perkin–Elmer TGA-7 instrument (Shelton, CT, USA). Samples (4–5 mg, duplicate) were heated from 30 °C to 800 °C at a heating rate of 20 °C/min under nitrogen. Data were analyzed using the Pyris v13.3 software.

#### 2.2.6. NMR Spectroscopy

PHBV samples were dissolved in CDCl_3_, and ^1^H- and ^13^C-nuclear magnetic resonance (NMR) spectra were recorded on an Advance Bruker 300 MHz and 500 MHZ spectrometer (Billerica, MA, USA) at room temperature, respectively. Spectra were analyzed using TopSpin V3.6.2 software (Billerica, MA, USA). The ^1^H-NMR spectra obtained for the PHBV samples were used to estimate monomer unit content. The molar fraction of the 3HV unit of the fractionated PHBV samples was estimated from the relative integrated CH_3_ (V5) resonance. ^13^C-NMR spectra was used to determine the sequence distribution of 3HV and 3HB in PHBV as previously described [8,28,29].

#### 2.2.7. Dynamic Rheology

Compression-molded discs (1.5 mm (h) × 25 mm Ø) were prepared from purified PHBV in a pellet die using a hydraulic press at room temperature. Complex viscosity (*η**) was measured as a function of angular frequency (*Ꞷ*) in the oscillatory mode at 180 °C using a Bohlin CVO 100 N rheometer (East Brunswick, NJ, USA) equipped with 25 mm diameter parallel plates, 0.5% strain, and the frequency range of 0.01 Hz to 100 Hz (*ω* = 0.0628 to 628 rad/s).

#### 2.2.8. Mechanical Properties

PHBV disc samples (0.3 g, vacuum dried at 60 °C overnight) were prepared in a pellet die using a hot press (PHI 300 mm × 300 mm) at 180 °C. PHBV discs were cut into 10 mm × 2.5 mm × 0.5 mm sized specimens for testing. Tensile testing was conducted with a gauge length of 10 mm and controlled force of 3 N/min until yield. 3-point bending tests were performed using a span of 10 mm with constant strain of 1% until yield. Experiments were performed on a DMA Q800 (TA instruments, New Castle, DE, USA) at room temperature with at least 5 replicates. Tensile strength, Young’s modulus, flexural strength, and flexural modulus were specified from the stress–strain curve using Universal Analysis v4.5A software (New Castle, DE, USA).

## 3. Results and Discussion

### 3.1. PHBV Extraction and Purification

PHBV containing biomass was produced at pilot-scale by MMC fed fermented dairy manure as described by Guho et al. [6]. The PHBV amount per dry cell of biomass, the 3HV content in the PHBV rich biomass, and the purity of PHBVs was determined by GC-MS and the results are given in Table 1. The 3HV content in purified PHBV was determined by ^1^H-NMR and data was from Guho et al. [6]. The lipid extracted from lyophilized biomass in the pre-treatment step (acetone wash) was 5–10% of the biomass weight which improves PHBV extraction [30]. The result showed that the 3HV molar fraction of the purified PHBV was less than that of PHBV rich biomass. The lower 3HV molar fraction in all the extracted polymers compared PHBV-rich biomass shows that CHCl_3_ selectively extracted PHBV with lower 3HV content. Wei et al. [8] reported the 3HV ratio of PHBV rich biomass produced using MMC was about 0.39 and decreased to 0.34 on purified PHBV. All the purified PHBVs showed lower HV content than PHBV rich biomass. This can be attributed to the effect of solvent type, extraction time and temperature on the HV content of the extracted PHBV [31]. There is an optimum time and temperature in extraction with every solvent and method. It has been seen that the HV content and *M_w_* of the extracted PHBVs decreased when extraction time exceeded the optimum level [31], and a longer extraction time can result in hydrolyses of polymer chains [32,33]. Furthermore, depending on the solvent type the atomic interactions between solvent and solute vary which affects the composition of the extracted polymer [19,34]. 

### 3.2. PHBV Molar Mass Determination 

A variety of parameters in both upstream and downstream processing of PHA production can influence the molar mass of the PHA produced. The upstream parameters include the microorganisms present in the MMC, the medium composition, the state of inoculum, and the fermentation conditions. The downstream processing parameters includes pre-treatment of biomass, extraction method, and subsequent polymer purification procedures [35]. The molar mass (*M_w_* and *M_n_*) and polydispersity index (*PDI*) of purified PHBV were determined by SEC-MALS and the results were from Guho et al. [6] (Table 2). The *PDI* varied between 1.06 and 1.43. The higher *PDI* indicates a broader range of distribution of molar mass. The *PDI* values of the PHBV in this study were lower than those reported for PHBV (*PDI* of 2.1–2.2) [8]. In an study PHBV extracted with CHCl_3_ showed a high *PDI* of 3.06 [36]. PHBV having low *PDI* can be used for applications such as controlled drug delivery [37]. Lower *PDI* values of the PHBV in this study can be attributed to both upstream and downstream parameters. Upstream processing parameters which mentioned earlier will determine the initial HV content and *M_w_* of the cultured PHBV. Downstream processing parameters include some polymer removal in the pretreatment step from lyophilized biomass, solvent type and its atomic interactions with solute, heating energy, and time applied in extraction and purification steps. The combination of these upstream and downstream parameters affects HV content, *M_w_*, *M_n_*, and as a result *PDI* of the purified polymer [6,8,19,31,32,33,34].

It was found that there was a negative linear correlation between *M_w_* (= −152.0 × HV + 42.68) and *M_n_* (= −134.27 × HV + 37.939) with the HV content of the extracted PHBVs (*R*^2^ ≥ 0.83). Pearson tests were performed between *M_w_* and HV content (*p* = 0.031), and *M_n_* and HV content (*p* = 0.014). Furthermore, similar correlation was seen between *M_w_* (= −33.261× HV + 17.475) with the HV content of the PHBV in lyophilized biomass (*R*^2^ = 0.91). Pearson test was performed between *M_w_* and HV content of the PHBV in lyophilized biomass (*p* = 0.012). There was no significant relation between *M_n_* with HV content, and PDI with HV content of the PHBVs in lyophilized biomass. 

### 3.3. FTIR Spectral Analysis

Purified PHBV samples were analyzed by FTIR spectroscopy to gain insight into its chemical structure; results are given in Table 3. The C–C stretching band occurs at 977 cm^−1^ [38,39]. The C–O stretching bands are observed between 1054 and 1129 cm^−1^ and from 1226 to 1275 cm^−1^ [38,39,40,41]. The band at 1179 cm^−1^ was assigned to the C–O–C stretching of the amorphous PHBV [42]. The band at 1720 cm^−1^ (highest intensity) was assigned to a C=O stretching of an ester structure in PHBV [6,38,40]. The C–H stretching bands of methyl (–CH_3_) and methylene (–CH_2_) groups appeared at 2975 cm^−1^ and 2933 cm^−1^, respectively [39,40,43]. Analysis of the FTIR spectra can be used to gain insight into the relative amounts of amorphous versus crystalline material in PHBV by determining the carbonyl index (*I_C=_**_O_*). Curve fitting of the crystalline band at 1720 cm^−1^ to amorphous band at 1740 cm^−1^ was used to determine band areas and *I_C=O_* (Figure 1) [25,26,31]. The *I_C=O_* for purified PHBV ranged from 2.01 to 2.59 (Table 4). A poor correlation (*R*^2^ = 0.53) between *I_C=O_* and *X_c_* by DSC (17–29%) was observed. These results are lower than reported values for PHB (*I_C=O_* 6.94 and *X_c_* 57.6%) [38], and higher than that reported for PHBV with a 25% 3HV content (*I_C=O_* 1.6 and *X_c_* 53.7%) [44]. It has been shown that as the crystallinity index decreased from 0.99 to 0.81 as HV content increased from 0% to 47% [45].

### 3.4. Thermal Analysis of PHBV

The purified PHBV samples and commercial standards were analyzed by DSC to determine their thermal properties (glass transition temperature, *T_g_*; melting temperature, *T_m_*; crystallization temperature (*T_c_*) and degree of crystallization, *X_c_*); the results obtained from Guho et al. [6] and the reanalyzed data are given in Table 4 and (Appendix A). The result of *T_g_* of the PHBV samples was similar to the results obtained from Guho et al. [6] (−0.7 °C to −7.4 °C vs. −2.8 °C to −10 °C) consistent with commercial PHBV (22% HV, −1.5 °C). The result of *T_m_*_1_ and *T_m_*_2_ of the PHBV samples was similar to the results obtained from Guho et al. [6], except PHBV-61 and -71 which showed one melting peak. Isomorphism phenomena can lead to the presence of two melting peaks [6,31]. As PHBV presents a semi-crystalline structure the crystals with higher HV content will have a higher amorphous phase ratio and thus during heating will be melted (*T_m_*_1_) first. In crystals with lower HV content, the crystallinity ratio is higher, so the crystals will melt at a higher temperature (*T_m_*_2_) [31,46]. Other factors that contribute to two melting points include the different crystalline morphologies (thickness, perfection, lamellar stability, or distribution), crystallites of different molar masses, physical aging, and amorphous phase relaxation [31,46].

There was no correlation between the melting temperature of the polymers and 3HV content. The *X_c_* of the purified PHBV obtained from first cooling cycle which was slightly higher than that obtained from Guho et al. [6] (16.6–29% vs. 13.9–25%). The *I_C=O_* of all the extracted PHBV samples ranged between 2.01 to 2.59 which was higher than the previously reported value of 0.949 for PHB [47]. PHB was shown to have a higher *X_c_* (61%), by X-ray diffraction, than PHBV (36.2%) and also showed higher value of *I_C=O_* by FTIR (3.8 vs. 2.7) [26]. The values of *I_C=O_* by FTIR and *X_c_* by DSC of the extracted PHBVs gave a poor correlation (*R*^2^ = 0.53). 

The *T_c_* for the extracted PHBVs ranged from 56 to 107 °C (Table 4). Sample PHBV-32 showed the highest *T_c_* (107.1 °C) with a 3HV content of 0.18 while sample PHBV-71 had the lowest *T_c_* (56.5 °C) and a higher 3HV content of 0.23. There was a negative linear correlation between *T_c_* and 3HV content (*R*^2^ = 0.54), and also between *T_c_* and *X_c_* of the extracted PHBVs (*R*^2^ = 0.54). An Increase in 3HV content results in an increase in the amorphous phase and thus leads to a reduction in *X_c_* and *T_c_* [48]. A lower *T_c_* indicates that the complete crystallization takes a longer time [49]. PHBV-15 has the lowest 3HV content (0.16), high *X_c_* (29%), and low *T_c_* of 62 °C. This can be attributed to the higher *PDI* of 1.43 in PHBV-15. Higher *PDI* indicates a larger size distribution of polymer chains. Shorter chains need less time for crystallization compared to longer chains which results in reduction of *T_c_* [49]. A *T_c_* value ranging from 41 to 82 °C for CHCl_3_ extracted PHBV with 3HV contents of (20–60%) have been reported [50]. Another study had shown that PHBV (16.6% HV) extracted with NaClO and CHCl_3_ had a *T_c_* value of 109 °C [51]. 

TGA analysis was performed to determine the thermal degradation behavior of the purified PHBV samples (Table 4, and Appendix A). All of the PHBV samples showed a 1-step degradation behavior. Based on the derivative thermogravimetric (DTG) curves the temperature at which the maximum weight loss occurred (*T_d_*) was determined. The *T_d_* values of the PHBV biopolymers was between 318 °C and 329 °C, which was higher than those reported in the literature for PHBV (*T_d_* = 286 °C) [52] and PHB (280 °C) [53]. For PHB thermal degradation occurs just above its *T_m_* which limits its processability [7]. The presence of 3HV in PHBV lowers its *T_m_* relative to *T_d_* and thus improves its processing window for molding and extrusion [21,31,49,54]. The thermal degradation of PHBV occurs in a single step due to a nonradical random chain scission process involving a six-membered ring transition state [24,31,52,55]. The onset temperature of degradation was from 286.4 °C to 303.7 °C, and the end set temperature of degradation was from 325.3 °C to 336.1 °C. These temperatures are higher than those reported in the other literatures [9,20]. The *T_onset_* of 177–238 °C, *T_d_* 276–296 °C, and end set temperature of 291–309 °C is reported for PHBV (5–20% HV) [9]. Furthermore, the *T_d1%_* (temperature at which 1% weight loss occurs) of 258 °C and *T_d_* 284 °C for PHB is reported [20] indicating an increase in thermal stability of PHBV samples studied here compared to pure PHB. While *T_d_* was found to decrease with increasing 3HV content (*R*^2^ = 0.49), the difference between *T_onset_* and *T_endset_* was found to increase with increasing 3HV content further reinforcing that the thermal stability of PHBV improved [7].

### 3.5. Comonomer and Composition Sequence Distribution Analysis

The molar HV content in PHBV was determined by ^1^H-NMR spectroscopy (Table 1, Appendix A). The molar HV content ranged between 0.16 and 0.24 which was in good agreement with those obtained by GC-MS (HV molar content 0.17 to 0.25, Table 1).

To determine whether the PHBV samples were random or co-block-polymer a sequence distribution (CSD) was determined by ^13^C-NMR spectroscopy (Figure 2) based on diad and triad sequence analysis [8]. The carbon resonances of HB and HV were split into multiplets due to diad and/or HV-centered triad comonomer sequences and assignments [8,29] (Figure 2b, and Appendix A). The carbonyl area was composed of four peaks at δ = 169.56, 169.38, 169.35, and 169.18 ppm attributed to different diad sequences of 3HV*3HV, 3HB*3HV, 3HV*3HB, and 3HB*3HB, respectively. The resonance of the 3HV side-chain methylene C (V4 and V2) was split into four triad sequences peaks attributed to 3HV-centered triads of 3HB-3HV*3HB, 3HV-3HV*3HB, 3HB-3HV*3HV, and 3HV-3HV*3HV. The chemical shifts observed (Appendix A) were in agreement with those reported previously [8,29].

The relative peak intensities obtained for the carbonyl (V1 and B1) resonances and the methylene (V2, B2, and V4) resonances of PHBV samples were used to determine the mole, diad, and HV-centered triad sequence distributions (Table 5). In the PHBV-15, -32, and -39 samples some signals were assigned to carbonyl and methylene C (B2, V2, V4) and in the PHBV-71 a signal assigned to methylene C (B2) were not observed. Therefore, the calculation of moles for diad and HV-centered triad fractions of HB and HV monomers on these samples were not performed.

The degree of randomness of copolymer based on experimental diad sequence distribution of the PHBV samples was determined by parameter *D* (Table 6), which is defined as
(2)$$D\;=\;\frac{F_{VV}F_{BB}}{F_{VB}F_{BV}}$$

*F_XY_* represents the mole fraction of *XY* sequence. The subscripts VV, BB, VB, and BV represent valerate-valerate, butyrate-butyrate, valerate-butyrate and butyrate-valerate of diad sequences, respectively. The statistically random copolymers have *D* value of 1.0. The *D* value for blocky and alternating copolymers are >1 and very close to 0, respectively. The term “blocky” means that the copolymer is a mixture of copolymers with different compositions or a mixture of PHB and PHV homopolymers. The parameter *D* is sensitive to the bimodal or multimodal (very large *D*) CSD, whereas it is not sensitive to the broadness of CSD [28].

A higher-order sequence distribution analysis such as the triad sequence distribution provides more accurate information on CSD. The degree of randomness in the PHBV copolymers based on the experimental triad sequence distribution was determined by the R parameter (Appendix A. In the PHBV copolymers with a completely randomized distribution of HV and HB monomers, *R* has a value of 1 (Table 6) [28].

The diad and triad sequence distribution of PHBV biopolymer samples were calculated using three models including (i) Bernoullian, (ii) first-order Markovian, and (iii) a mixture of two Bernoullian random copolymers (explanation of the models is given in Appendix A [8,29]. The result of molar fractions of diad, and 3HV-centered triads of 3HB and 3HV for PHBV-46, -61, -78, and -84 is listed in Table 7.

The PHBV samples sequence distributions followed the Bernoullian statistics model. The results confirm that the PHBV samples were all random copolymers and these results are in agreement with those reported previously [8,29].

### 3.6. Rheology

Rheology presents information about of the behavior of polymer melts through measuring the flow (viscosity) as a function of shear rate [56]. From dynamic rheological measurements the complex viscosity (*η**) was determined for PHBV at 180 °C against shear rate (*Ꞷ*) as frequency sweeps (also known as flow curves); results are shown in Figure 3. The PHBV flow curves show a reduction in *η** with *Ꞷ* showing shear thinning behavior typical for non-Newtonian fluid [31,57]. At high shear rates the anisotropic polymer chains with high molar mass will disentangle and aligned along the shear direction [57,58]. Thus, less intermolecular interaction happens among the polymer chains resulting in higher free volume and a lower viscosity [31,57,58]. The *η** of PHBV-15 decreased more rapidly than the other PHBV samples (Figure 3 and Table 8). This might be attributed to the lower purity (80%) of the polymer and partial plasticizing effect of impurities present and high *PDI* of 1.47 in the polymer [31,59,60]. Smaller chains act as a plasticizer for longer chains, increasing the free volume space between polymer chains leading to improvement in the mobility of the polymer chains. The viscosity of the polymer therefore decreases with higher slope comparing the polymers without plasticizer [60].

To quantify the shear thinning behavior, the rheological data were fitted to the Power–law (Ostwald–de Waele) model, which represents viscosity as a function of some power of the shear rate (*Ꞷ*) [31,61]. The Cox–Merz rule is used to establish the relationship between steady shear and dynamic shear viscosity, which is valid for PHA, PHB, and PHBV [31,61]. Exchanging the steady shear terms with the dynamic viscosity terms yields a modified Power-law model, expressed as:
(3)$$|\eta *(Ꞷ)|\;=\;K{\left(Ꞷ\right)}^{n-1}$$

The modified model has two parameters of *K* and *n* that must be fitted to experimental data. These parameters are determined from the plot double log *η**-*Ꞷ*. The parameter *K* is known as the consistency coefficient, is the y-intercept of the log *η**-log *Ꞷ* fitted plot. The parameter *n* is the non-Newtonian or flow behavior index which is the slope of the log *η**-log *Ꞷ* fitted curve. For *n* < 1 the material is pseudo-plastic, when *n* = 1 the material is Newtonian, and *n >* 1 the material is dilatant [31,61,62,63]. However, a drawback to this model is that the power-law model does not accurately represent experimental data in the low and high shear ranges, so it should not be extended to these regions [62,63]. The goodness of fit for the models *R*^2^ was used. All the PHBV samples had *R*^2^ ≥ 0.92. The *K* and *n* parameters, and complex viscosity at the frequency of *Ꞷ* = 6.28 rad·s^−1^ (1 Hz) for PHBV-15 to -84 are summarized in Table 8. The Power-law model was well fitted to the data, with *K* ranging from 3.27 to 28.07 Pa·s. The *n* ranged from 0.645 to 0.888 showing pseudo-plastic behavior. Reported values for PHBV (5% HV) at 160 °C at were *K* = 1.35 × 10^6^ Pa·s and *n* = 0.0823 [58,61]. The *η** of the PHBV samples at 1 Hz ranged from 1.7 Pa·s to 14.57 Pa·s. The PHBV samples with higher HV content showed higher *η** at 1 Hz. It is expected an increase in the HV content leads to a reduction in *η** [31,64,65]. This difference can be attributed to the simultaneous effect of higher crystallinity which increases the *η** and lower *PDI* which decreases the *η** in polymers [60].

### 3.7. Mechanical Properties

The mechanical (flexural and tensile) properties of materials provide information on how they will behave under load. Table 9 shows the tensile and flexural properties of the PHBV samples and a commercial PHBV (8% HV) sample. The PHBV tensile strength ranged between 10.1 and 13.7 MPa and Young’s moduli ranged between 0.57 and 1.12 GPa. The commercial PHBV had a strength of 25.9 MPa and modulus 0.75 GPa. Tensile strength values from the literature were 27.1 MPa (injection molded 8% HV) [66], 29.14 MPa (solution cast 5% HV) [67], and 14.1 MPa (solution cast 12% HV) [68]. While the Young’s modulus values were 3.5 GPa for injection molded (8% HV) [66], 2.19 GPa (solution cast 5% HV) [67], and 0.82 GPa (solution cast 12% HV) [68]. In a separate study, the tensile strength and Young’s modulus of 20 MPa and 0.8 GPa for PHBV-20% HV were reported [69]. An inverse relationship was between strength and modulus values and HV content.

The PHBV samples were shown to have a flexural strength of 18.1 to 27.5 MPa and moduli of 0.80 to 1.54 GPa. Literature values for flexural strength for PHBV(8% HV) were 34.5 MPa [66] and 28.2 MPa [70] and the flexural moduli for PHBV(8%HV) were 3.2 GPa [66] and 1.3 GPa [70]. The low mechanical properties for the pilot plant PHBV samples were due to their high HV content. A negative linear relationship between flexural strength and HV% (*R*^2^ = 0.68), flexural modulus and HV% (*R*^2^ = 0.78), tensile strength and HV% (*R*^2^ = 0.69), and Young’s modulus and HV% (*R*^2^ = 0.83) were observed. The strength and flexibility of these polymers was comparable to LDPE which makes them applicable for the packaging industry [5,8]. It has been reported that an increase in HV content (0 to 30%) reduces the melting point, crystallization rate, tensile strength, and stiffness of the PHBV [71]. PHBV shows a comparable modulus to polyethylene terephthalate (PET), higher than to that for low density polyethylene (LDPE) and polypropylene (PP) [71]. For comparative purposes, the tensile strength and Young’s moduli for PHB were 40 MPa and 3.5 GPa, for LDPE were 10 MPa and 0.2 GPa, and PP were 38 MPa and 1.7 GPa PP [69,72]. Low content HV PHBV has applications where rigidity is required such as bottle caps, while high content HV PHBV would be suitable as flexible packaging and bottles [71].

The molar mass distribution of polymers, PDI, is known to influence their properties. Polymers with higher PDI contain some shorter chain polymers that can act as a plasticizer and therefore reduce their mechanical properties [60]. A negative correlation between flexural strength and PDI (*R*^2^ = 0.68), flexural modulus and PDI (*R*^2^ = 0.78), tensile strength and PDI (*R*^2^ = 0.69), and Young’s modulus and PDI (*R*^2^ = 0.83) were observed.

## 4. Conclusions

The successful investigation of large lab-scale Soxhlet extraction using CHCl_3_, recovery, and characterization of properties of PHBV with various copolymer compositions biosynthesized by mixed microbial consortia fed fermented dairy manure was performed. The purified PHBV showed a higher range of thermal properties, broader processing window, and more flexible mechanical properties as compared to commercial PHBV (control), which make the produced PHBV suitable for processing applications such as blow and injection molding. Generally, the purified PHBV showed some variations in properties over the 3 months of operation, which may be attributable to as-yet-refined operational strategies and process controls. To reduce variations at scale would require the operations to be carefully monitored and process controlled in future studies.

## Figures and Tables

**Figure 1 polymers-14-04140-f001:**
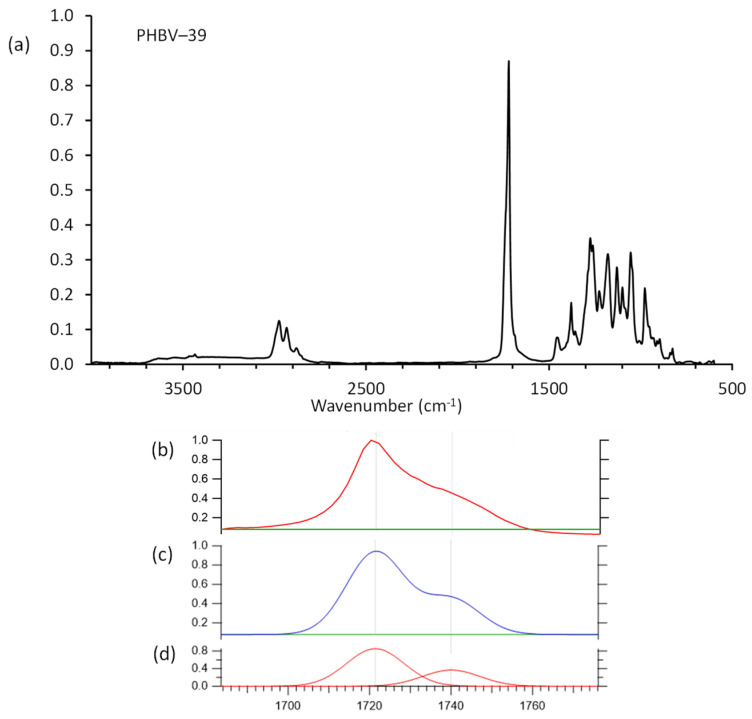
(**a**) FTIR spectrum for purified PHBV-39 sample; (**b**) expanded spectrum showing the carbonyl region before fitting; (**c**) expanded spectrum showing the carbonyl region after fitting; (**d**) fitted carbonyl bands.

**Figure 2 polymers-14-04140-f002:**
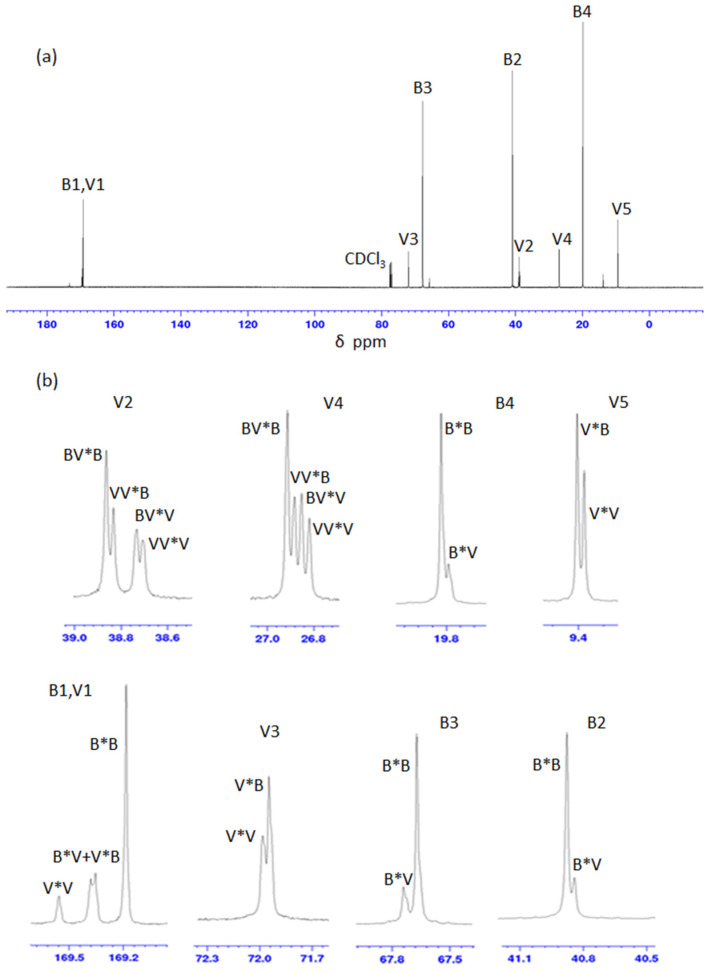
^13^C-NMR (**a**) full spectrum; (**b**) expanded region associating to the splitting of individual resonances of PHBV-61. * denotes linkage between units V and B.

**Figure 3 polymers-14-04140-f003:**
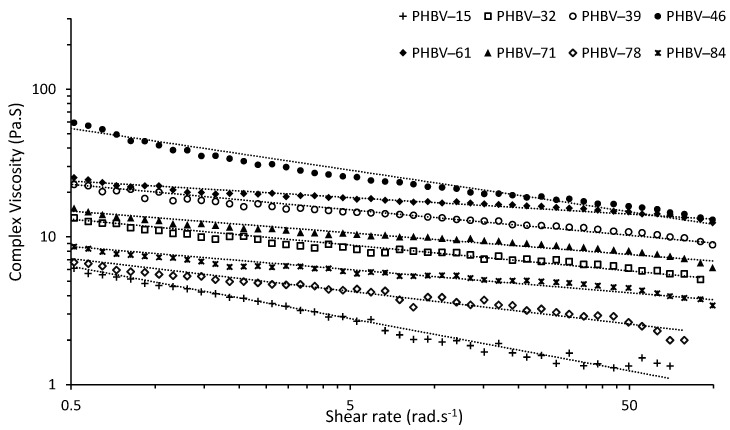
Complex viscosity (*η**) versus shear rate (*Ꞷ*) plots for PHBV samples (180 °C) and power-law fitted models (dotted lines).

**Table 1 polymers-14-04140-t001:** PHBV and 3HV content in biomass, crude and purified PHBV yields and 3HV content in purified PHBV.

Operational Day	Lyophilized Biomass		PHBV Yield (%)		Purified PHBV		
	PHBV (%)	*f_3HV_* ^a^	Crude ^b^	Pure ^c^	Purity (%)	*f_3HV_* ^a^	PHBV (%)
	GC-MS	GC-MS				GC-MS	^1^H-NMR [6]
15	11.6 ± 0.1	0.23 ± 0.01	20.6	8.9	80.6 ± 3.0	0.17 ± 0.01	0.16 ± 0.02
32	24.5 ± 0.4	0.27 ± 0.00	31.2	14.6	81.7 ± 2.1	0.19 ± 0.01	0.18 ± 0.00
39	35.6 ± 1.9	0.32 ± 0.00	31.8	18.6	88.5 ± 4.7	0.21 ± 0.01	0.19 ± 0.00
46	32.7 ± 0.8	0.26 ± 0.00	27.7	17.5	91.2 ± 0.6	0.25 ± 0.00	0.23 ± 0.00
61	31.1 ± 4.3	0.29 ± 0.03	26.0	13.3	91.2 ± 1.1	0.25 ± 0.00	0.24 ± 0.00
71	31.1 ± 0.3	0.31 ± 0.00	29.0	18.1	87.6 ± 4.4	0.25 ± 0.01	0.23 ± 0.00
78	30.5 ± 0.8	0.28 ± 0.00	26.5	20.4	95.6 ± 4.9	0.23 ± 0.01	0.22 ± 0.00
84	31.1 ± 1.3	0.28 ± 0.00	NA	26.9	95.7 ± 3.9	0.23 ± 0.01	0.21 ± 0.00

^a^ *f_3HV_* is the molar fraction of 3HV in PHBV. ^b^ Crude Yield (%) = (wt _crude PHBV_/wt _Ash free lyophilized biomass_) × 100. ^c^ Pure Yield (%) = (wt _pure PHBV_ × PHBV purity/wt _Ash free lyophilized biomass_) × 100.

**Table 2 polymers-14-04140-t002:** Molar Mass and Poly dispersity index of purified PHBV obtained from CHCl_3_ extraction of biomass ^a,b^.

Sample	Molar Mass ^a,b^ [6]		
	*M_w_*	*M_n_*	*PDI*
	(×10^5^ g·mol^−1^)	(×10^5^ g·mol^−1^)	
PHBV-15	28.0 ± 0.43	19.6 ± 0.5	1.43 ± 0.04
PHBV-32	8.67 ± 0.31	7.98 ± 0.33	1.09 ± 0.06
PHBV-39	6.96 ± 0.38	6.45 ± 0.32	1.08 ± 0.08
PHBV-46	8.54 ± 0.18	7.52 ± 0.09	1.14 ± 0.03
PHBV-61	6.49 ± 0.18	5.99 ± 0.21	1.08 ± 0.05
PHBV-71	6.86 ± 0.49	6.40 ± 0.39	1.07 ± 0.1
PHBV-78	8.45 ± 0.16	7.96 ± 0.15	1.06 ± 0.03
PHBV-84	11.3 ± 0.25	10.1 ± 0.21	1.12 ± 0.03

^a^ Symbols are defined as: *M_w_* is the weight-average molecular weight; *M_n_* is the number-average molecular weight; *PDI* is the polydispersity index (*PDI =*
*M_w_/M_n_*). ^b^ Values are mean ± standard deviation. The standard deviation has been rounded up to the mean’s reported precision wherever necessary.

**Table 3 polymers-14-04140-t003:** Assignments of PHBV IR bands.

Wavenumber (cm^−1^)	Correspondence	Reference
977	C–C backbone stretching vibration of crystalline PHBV	[38,39]
1054	O–C–C stretching	[40]
1099	asymmetric O–C–C stretching	[41]
1129	symmetric C–O–C stretching of amorphous PHBV	[41]
1179	asymmetric C–O–C stretching of amorphous PHBV	[42]
1226	symmetric C–O stretching of crystalline PHBV	[38]
1261	symmetric C–O stretching of aliphatic esters	[41]
1275	symmetric C–O stretching of carbonyl group of crystalline PHBV	[39,40,43]
1379	symmetric C–H bending vibration of methyl groups	[40]
1452	asymmetric C–H stretching and bending vibrations of methyl and methylene groups	[39,40,43]
1720	symmetric C=O stretching of crystalline PHBV	[6,38,40]
2933	symmetric vibration of C–H of methylene groups	[39,40,43]
2975	asymmetric vibration of C–H of methyl groups	[39,40,43]

**Table 4 polymers-14-04140-t004:** Physical and thermal properties of purified PHBV obtained from CHCl_3_ extraction of biomass.

Sample		DSC				FTIR	TGA		
	*T_g_*	*T_m_* _1_	*T_m_* _2_	*T_C_*	*X_c_*	*I_C=O_*	*T_onset_*	*T_d_*	*T_endset_*
	(°C)	(°C)	(°C)	(°C)	(%)		(°C)	(°C)	(°C)
PHBV-15	−0.7	126.1	152.5	61.8	29.0	2.48	286	318	325
PHBV-32	−3.5	150.9	166.5	107.1	19.3	2.18	304	329	336
PHBV-39	−5.3	159.7	170.1	104.8	25.8	2.35	299	325	333
PHBV-46	−7.4	153.8	165.3	80.0	21.6	2.25	297	325	331
PHBV-61	−7.2	-	162.6	71.5	23.8	2.20	298	327	333
PHBV-71	−2.5	-	155.6	56.5	16.6	2.28	295	322	332
PHBV-78	−1.6	128.6	152.1	64.2	23.3	2.01	289	320	327
PHBV-84	−1.9	130.9	153.1	66.4	24.3	2.59	294	326	331

**Table 5 polymers-14-04140-t005:** Experimental diad and 3HV-centered triad relative peak intensities for purified PHBV from different operational days.

Sample	Carbon	HV (mol·mol^−1^) ^a^	3HV	3HB	3HV*3HV	3HV*3HB	3HB*3HV	3HB*3HB	3HV-3HV*3HV	3HB-3HV*3HV	3HV-3HV*3HB	3HB-3HV*3HB
PHBV-46	V1,B1		0.228	0.772	0.080	0.148	0.138	0.634				
	B2			0.776			0.153	0.623				
	V2		0.224		0.078	0.146			0.150	0.200	0.252	0.398
	V4								0.160	0.227	0.230	0.383
	av	0.23	0.226	0.774	0.079	0.147	0.146	0.628	0.035	0.048	0.054	0.088
PHBV-61	V1,B1		0.219	0.781	0.078	0.141	0.125	0.657				
	B2			0.779			0.141	0.639				
	V2		0.221		0.077	0.144			0.159	0.189	0.248	0.405
	V4								0.170	0.222	0.215	0.394
PHBV-78	av	0.240	0.220	0.780	0.077	0.143	0.133	0.648	0.036	0.045	0.051	0.088
	V1,B1		0.239	0.768	0.086	0.154	0.146	0.621				
	B2			0.774			0.157	0.617				
	V2		0.226		0.079	0.147			0.165	0.183	0.226	0.741
	V4								0.166	0.207	0.216	0.411
	av	0.220	0.232	0.771	0.082	0.150	0.152	0.619	0.039	0.045	0.051	0.134
PHBV-84	V1,B1		0.197	0.803	0.066	0.131	0.117	0.686				
	B2			0.798			0.125	0.674				
	V2		0.202		0.070	0.132			0.173	0.172	0.223	0.432
	V4								0.184	0.200	0.198	0.418
	av	0.210	0.199	0.801	0.068	0.132	0.121	0.680	0.036	0.037	0.042	0.085

^a^ Molar fraction of 3HV is determined by ^1^H-NMR spectroscopy.

**Table 6 polymers-14-04140-t006:** Parameters *D*, *R*, experimental number average sequence lengths of HV units (*L_V_^E^*), number average sequence length of randomly distributed HV units in copolymer (*L_V_^R^*), ratio between the concentration of HV and HB units (*k*), four conditional probabilities (*Pij’s*) and the reaction index (*r_1_r_2_*) for PHBV operational days 46, 61, 78 and 84.

Sample	*D*	*R*	*L_V_^R^*	*L_V_^E^*	*K*	*P_VV_* ^a^	*P_VB_* ^a^	*P_BV_* ^a^	*P_BB_* ^a^	*r* _1_ *r* _2_ ^b^	*A*	*B*	*X*	Model ^c^	Sequence Distribution
PHBV-46	1	1	1.29	1.29	3.43									exptl	Random
														(i)	
						0.35	0.65	0.19	0.81	2.33				(ii)	
											0.41	0.03	0.51	(iii)	
PHBV-61	1	1	1.28	1.28	3.55									exptl	Random
														(i)	
						0.35	0.65	0.17	0.83	2.64				(ii)	
											0.06	0.44	0.57	(iii)	
PHBV-78	1	1	1.30	1.30	3.32									exptl	Random
														(i)	
						0.35	0.65	0.20	0.80	2.23				(ii)	
											0.24	0.823	0.95	(iii)	
PHBV-84	1	1	1.25	1.25	4.02									exptl	Random
														(i)	
						0.34	0.66	0.15	0.85	2.88				(ii)	
											0.08	0.5	0.72	(iii)	

^a^ The estimated errors in the values of *Pij’s* are < 0.005. ^b^ *r*_1_
*= P_BB_/P_BV_* and *r*_2_
*= P_VV_/P_VB_*_._ ^c^ exptl represent experimental data; (i), (ii), and (iii) are calculated values by Bernoullian model, first-order Markovian model, and mixture of two Bernoullian random copolymers model, respectively.

**Table 7 polymers-14-04140-t007:** Experimental and calculated mole fractions of diad, and 3HV-centered triad sequence distributions of PHBV −46, −61, −78, and −84.

Sample	Model ^a^	*F_V_* ^b^	*F_V_* ^c^	*F_B_* ^c^	*F_VV_* ^c^	*F_VB/BV_* ^c^	*F_BB_* ^c^	*F_VVV_* ^c^	*F_BVV/VVB_* ^c^	*F_BVB_* ^c^
PHBV −46	exptl	0.23	0.23	0.77	0.08	1.01	0.63	0.03	0.89	0.09
	(i)		0.23	0.77	0.05	0.17	0.60	0.01	0.04	0.14
	(ii)		0.22	0.78	0.08	0.15	0.63	0.03	0.05	0.09
	(iii)		0.22	0.78	0.09	0.14	0.64	0.03	0.05	0.09
PHBV −61	exptl	0.24	0.22	0.78	0.08	1.07	0.65	0.04	0.89	0.09
	(i)		0.22	0.78	0.05	0.17	0.61	0.01	0.04	0.13
	(ii)		0.21	0.79	0.07	0.13	0.66	0.03	0.05	0.09
	(iii)		0.22	0.78	0.09	0.14	0.64	0.04	0.05	0.09
PHBV −78	exptl	0.22	0.23	0.77	0.08	0.99	0.62	0.04	0.88	0.13
	(i)		0.23	0.77	0.05	0.18	0.59	0.01	0.04	0.14
	(ii)		0.23	0.77	0.08	0.15	0.62	0.03	0.05	0.10
	(iii)		0.27	0.73	0.09	0.18	0.55	0.04	0.05	0.13
PHBV-84	exptl	0.21	0.20	0.80	0.07	1.09	0.68	0.04	0.88	0.08
	(i)		0.20	0.80	0.04	0.16	0.64	0.01	0.03	0.13
	(ii)		0.19	0.81	0.06	0.12	0.69	0.02	0.04	0.08
	(iii)		0.20	0.80	0.07	0.12	0.68	0.04	0.04	0.08

^a^ exptl represent experimental data; (i), (ii), and (iii) are calculated values by Bernoullian model, first-order Markovian model, and mixture of two Bernoullian random copolymers model, respectively. ^b^ 3HV molar fraction (mol·mol^−1^) determined by ^1^H-NMR spectroscopy. ^c^ *F_X_*, *F_XY_*, and *F_XVY_* represent mole fractions of sequence *X, XY*, and *XVY*, respectively, where *X, Y = V* or *B*.

**Table 8 polymers-14-04140-t008:** Power–Law parameters (*K* and *n*) and complex viscosity (*η**) at *Ꞷ* = 6.28 rad·s^−1^ (1 Hz) determined at 180 °C for the purified PHBV samples.

Sample	*K* (Pa·s)	*n*	*η** (Pa·s)	*R* ^2^
PHBV-15	3.27	0.645	1.7	0.982
PHBV-32	9.12	0.824	6.6	0.940
PHBV-39	15.48	0.827	11.26	0.931
PHBV-46	28.07	0.717	16.7	0.921
PHBV-61	17.91	0.888	14.57	0.951
PHBV-71	10.43	0.853	7.95	0.945
PHBV-78	4.71	0.778	3.13	0.945
PHBV-84	6.19	0.843	4.64	0.938

**Table 9 polymers-14-04140-t009:** Tensile and flexural properties of purified PHBV from different operational days.

PHBV Sample	Tensile Strength	Young’s Modulus	Flexural Strength	Flexural Modulus
	MPa	GPa	MPa	GPa
PHBV-15	12.3 ± 1.0 ^bc^	1.12 ± 0.21 ^a^	27.5 ± 1.5 ^b^	1.54 ± 0.13 ^b^
PHBV-32	13.7 ± 1.1 ^b^	0.83 ± 0.1 ^b^	27.4 ± 1.1 ^b^	1.27 ± 0.06 ^c^
PHBV-39	12.7 ± 1.9 ^bc^	0.87 ± 0.04 ^b^	20.8 ± 1.4 ^c^	1.02 ± 0.08 ^d^
PHBV-46	12.2 ± 1.8 ^bc^	0.57 ± 0.07 ^c^	19.8 ± 1.6 ^cd^	0.82 ± 0.06 ^e^
PHBV-61	13.3 ± 0.3 ^b^	0.61 ± 0.06 ^c^	18.1 ± 0.6 ^d^	0.80 ± 0.06 ^e^
PHBV-71	10.1 ± 1.4 ^c^	0.58 ± 0.05 ^c^	20.3 ± 1.4 ^cd^	0.86 ± 0.06 ^de^
PHBV-78	13.2 ± 0.7 ^b^	0.67 ± 0.03 ^bc^	21.2 ± 1.5 ^c^	0.95 ± 0.10 ^de^
PHBV-84	11.7 ± 0.9 ^bc^	0.79 ± 0.16 ^bc^	27.3 ± 0.8 ^b^	1.31 ± 0.03 ^c^
Std PHBV-8%HV	25.9 ± 2.2 ^a^	0.75 ± 0.09 ^bc^	59.2 ± 0.8 ^a^	4.65 ± 0.15 ^a^

^a^ Values are mean ± standard deviation. Where necessary, the standard deviation has been rounded up to the mean’s reported precision. ^b^ Statistically difference of mean of results is measured via Tukey test (*p*-value < 0.05) and showed by superscript letters (a, b, c, d, e).

## Data Availability

The data presented in this study are available on request from the corresponding author.

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
