# Peer review of "Effect of 3-Hydroxyvalerate Content on Thermal, Mechanical, and Rheological Properties of Poly(3-hydroxybutyrate-co-3-hydroxyvalerate) Biopolymers Produced from Fermented Dairy Manure"

_polymers, 2022, doi:10.3390/polym14194140_

Round 1

Reviewer 1 Report

Effect of HV content on Thermal, Mechanical, and Rheological properties of PHBV biopolymers produced from fermented dairy manure by

Maryam Abbasi, Dikshya Pokhrel, Erik R. Coats, Nicholas M. Guho and Armando G. McDonald

Polymers 2022, 14, x. https://doi.org/10.3390/xxxxx

Dear authors,

I have read your paper and have some comments, which will help in proper revision of your paper. Please consider the following:

1.    Page 1, Title – please include the “HV” explanation, as not every reader should know what it means.

2.    Page 1, abstract, line 17 – please include “3HB” explanation.

3.    Page 3, section 2.2.1. line 103 – please include explanation for the samples code: “e.g., PHBV-15 – PHBV at 15th operational day (for example.

4.    Page 3, section 2.2.4. FTIR spectroscopy – why only 2 measurements have been performed for these materials. In ATR mode it is normal to take at least 5 measurements per sample and average values present. What was the type of normalization? Please provide more information on the curve fitting procedure, for example what was the type of signal Gaussian or other. It seems that the results of curve fitting may be wrong as too less measurements have been performed and the curve fitting procedure refer to only 2 peaks in the region of carbonyl vibration of ca. 1700-1760 cm-1. Please consider to repeat the measurements, averaging, normalization and curve fitting procedure. Remember as well, that those values may differ form DSC data, as it is completely different method. Moreover, ATR mode is less sensible itself, as the measurement strongly related to the power of sample press to the crystal (the contact with the crystal).

5.    Page 3, section 2.2.5. Thermal analysis – please provide more information about the measurements, i.e. the type of the crucible, the gas flow, calibration of the instrument. Averaging is ok for those results, however evaluation of standard deviation is a serious mistake, as 2 values for such evaluation is too less.

Please provide DSC (showing both heating scans) and TG/DTG curves.

Please correct the eq.1, H values should be italic, as well as “X”. The same for the rest values measured, as molecular weight “M”, temperature “T”.

6.    Page 4, section 2.2.8. Mechanical properties – please provide if possible the information about the standard for tensile testing.

7.    Page 5, lines 211-217 – please correct the sentence as it is hardly to understand.

8.    Page 6, Table 2 – please consider to divide this table in two, as the table is difficult to read. The standard deviation for DSC and TG is impossible to calculate as only 2 values have been determined for each parameter.

9.    Page 6, section FTIR spectra analysis, line 236 – “Purified PHBV samples were analyzed by FTIR spectroscopy to gain insight into its chemical structure and band assignments; results are given in Table 3.” – please correct, IR analysis allow to investigate functional groups, not the band assignments. The band assignment comes from the researcher’ analysis.

A poor correlation between IR and DSC results is quite normal.

10. Page 7, Table 3 – please correct the clearness of the table, it is difficult to read. Please correct: 2933 is assigned to the methylene group vibration. Please provide explanation for the fact that PHBV-61 did not show methylene bands.

11. Page 8, Fig. 1 – please show the Y axis for IR and correct the Fig.1b to be more clear, indicate the raw curve and curve after fitting, and the “0” and “1” assignments. It seems that more than 2 peaks are present in this region, please correct the curve fitting procedure.

12. Page 8, section 3.4. Thermal analysis of PHBV – line 265-270 – please provide some comment on the one/two melting peaks observed for these materials. What it means? Line 273-274 “The values of IC=O by FTIR and Xc by DSC of the extracted PHBVs gave a poor correlation (R2= 0.53).” – those values are different than those from page 6.

13. Page 9, eq.2 – please provide the explanation for “F” values, is this a fraction value?

14. Page 11, table 4 – please correct the table to be more clear.

15. Page 12, Table 5 – please provide the explanation for “K”, “P” and “A, B, X”. You can omit the last column of “sequence distribution” as it is the same for all cases, and put this information in the text.

16. Page 12, table 6 – please provide explanation for “F” values.

17. Page 15, section 3.7. Mechanical properties – it seems that pilot plant PHBV samples showed lower values of mechanical parameters, as compared to the standard. Please correct the section, especially lines 444-456.

18. Page 15, Conclusions, line 467 “…and improved mechanical properties as compared to commercial PHBV (control)..” – they are lower, not improved, please correct.

Author Response

I have read your paper and have some comments, which will help in proper revision of your paper. Please consider the following:

  1. Page 1, Title – please include the “HV” explanation, as not every reader should know what it means.

We corrected the title and added the explanation of HV and PHBV to it. The updated title is as follows:

Effect of 3-hydroxyvalerate content on Thermal, Mechanical, and Rheological properties of Poly (3-hydroxybutyrate-co-3-hydroxyvalerate) biopolymers produced from fermented dairy manure”

  1. Page 1, abstract, line 17 – please include “3HB” explanation.

We added an explanation of HB to Line 17 in the abstract as the following:

13C-NMR spectroscopy revealed that the extracted PHBVs have a random distribution of 3HV and 3-hydroxybutyrate (3HB) units with 3HV content between 16% and 24%.”

  1. Page 3, section 2.2.1. line 103 – please include explanation for the samples code: “e.g., PHBV-15 – PHBV at 15th operational day (for example).

We have corrected and changed the samples codes, and added explanations as the following:

The sample code name for purified PHBV obtained at an operational day will coded PHBV-xx (e.g., PHBV-15 – PHBV at 15th operational day).”

  1. Page 3, section 2.2.4. FTIR spectroscopy – why only 2 measurements have been performed for these materials. In ATR mode it is normal to take at least 5 measurements per sample and average values present. What was the type of normalization? Please provide more information on the curve fitting procedure, for example what was the type of signal Gaussian or other. It seems that the results of curve fitting may be wrong as too less measurements have been performed and the curve fitting procedure refer to only 2 peaks in the region of carbonyl vibration of ca. 1700-1760 cm-1. Please consider to repeat the measurements, averaging, normalization and curve fitting procedure. Remember as well, that those values may differ form DSC data, as it is completely different method. Moreover, ATR mode is less sensible itself, as the measurement strongly related to the power of sample press to the crystal (the contact with the crystal).
  • Unfortunately, we do not have more samples to redo the FTIR experiments and to analyze the data. Also, we do not have enough time to do any more experiments because of the time allocated to submitting a revised manuscript (10 days).
  • We believe the duplicate analysis is sufficient since the purified PHBV was homogeneous.
  • We first averaged the data, corrected the data against the baseline (wavenumbers/X-axis), then normalized it against the absorbance (Y) axis to unit one. Then, we used the following data for curve fitting using Igor pro software and the Gaussian function for peak fitting and a peak width at half height of 10 cm-1 to evaluate the crystallinity of the sample through the carbonyl vibration appearing from 1680 to 1780 cm-1.
  • We performed the same method on all of the samples, the pressure applied (spring loaded clutch on accessory) to all samples was the same.

We have modified the text in the methods as: “The carbonyl index (IC=O = A1720/A1740) was determined on normalized spectra that were curve-fitted using Igor Pro 9 software (WaveMetrics) [25,26]. The area (A) of each carbonyl band at 1720 cm-1 and 1740 cm-1 were Gaussian curve fitted using a peak width at half height of 10 cm-1 and integrated

  1. Page 3, section 2.2.5. Thermal analysis – please provide more information about the measurements, i.e. the type of the crucible, the gas flow, calibration of the instrument. Averaging is ok for those results, however evaluation of standard deviation is a serious mistake, as 2 values for such evaluation is too less.

Please provide DSC (showing both heating scans) and TG/DTG curves.

We used Tzero aluminum pans (from TA instruments) for the measurements. For calibration we followed the manufacturer's guidelines and calibrated the DSC with indium and sapphire standards. The N2 flowrate is set by the instrument at 50 mL/min. We believe duplicate analyses is sufficient and we gave an average result only and removed the standard deviation in Table 3. We have used all the samples and have none left for further analyses. Also, we do not have enough time do any more experiments because of the time allocated to submitting a revised manuscript (10 days).

We have added text as follows:

Differential scanning calorimetry (DSC) was performed on purified PHBV (4-6 mg, in duplicate using TzeroTM aluminum pans) using a TA instrument model Q200 DSC with refrigerated cooling under nitrogen (50 mL/min). The DSC was calibrated using indium and sapphire standards following manufacturer protocols.

We have provided TG and DTG curves from the TGA test (Figure S2) and heating scans from the DSC test in the supplementary material (Figure S1).

Please correct the eq.1, H values should be italic, as well as “X”. The same for the rest values measured, as molecular weight “M”, temperature “T”.

We applied the corrections throughout the manuscript.

  1. Page 4, section 2.2.8. Mechanical properties – please provide if possible, the information about the standard for tensile testing.

We followed the methods of the following paper and TA instruments application note (TA038) for tensile testing using DMA. We followed the manufacturers application note for doing tensile testing using the DMA.

Wei, L., Guho, N. M., Coats, E. R., & McDonald, A. G. (2014). Characterization of poly (3‐hydroxybutyrate‐co‐3‐hydroxyvalerate) biosynthesized by mixed microbial consortia fed fermented dairy manure. Journal of Applied Polymer Science131(11).

TA Instruments applications note TA-038. https://www.tainstruments.com/pdf/literature/TA038.pdf . Accessed 09-22-2022.

  1. Page 5, lines 211-217 – please correct the sentence as it is hardly to understand.

Thank you for the comment. We rewrote the mentioned lines as the following:

Upstream processing parameters which mentioned earlier will determine the initial HV content and Mw of the cultured PHBV. Downstream processing parameters include some polymer removal in the pretreatment step from lyophilized biomass, solvent type and its atomic interactions with solute, heating energy, and time applied in extraction and purification steps. The combination of these upstream and downstream parameters affects HV content, Mw, Mn, and as a result PDI of the purified polymer [6,8,19,31-34].”

  1. Page 6, Table 2 – please consider to divide this table in two, as the table is difficult to read. The standard deviation for DSC and TG is impossible to calculate as only 2 values have been determined for each parameter.

We have divided table 2 into two tables “2 and 3”. we gave an average result only and removed the standard deviation.

  1. Page 6, section FTIR spectra analysis, line 236 – “Purified PHBV samples were analyzed by FTIR spectroscopy to gain insight into its chemical structure and band assignments; results are given in Table 3.” – please correct, IR analysis allow to investigate functional groups, not the band assignments. The band assignment comes from the researcher’ analysis.

A poor correlation between IR and DSC results is quite normal.

We removed the “and band assignments” from line 236 as the following:

Purified PHBV samples were analyzed by FTIR spectroscopy to gain insight into its chemical structure; results are given in Table 4.”

  1. Page 7, Table 3 – please correct the clearness of the table, it is difficult to read. Please correct: 2933 is assigned to the methylene group vibration. Please provide explanation for the fact that PHBV-61 did not show methylene bands.

All the tables are imported from excel and unfortunately, there is no way to make them clearer.

The band assignment at 2933 was corrected.

PHBV-61 did show methylene bands. We corrected that in the manuscript.

  1. Page 8, Fig. 1 – please show the Y axis for IR and correct the Fig.1b to be more clear, indicate the raw curve and curve after fitting, and the “0” and “1” assignments. It seems that more than 2 peaks are present in this region, please correct the curve fitting procedure.

We applied the requested edits on the figure1 and separated them in (b), (c) and (d). We re-performed the fitting. Two peaks appeared after fitting and the result did not change.

  1. Page 8, section 3.4. Thermal analysis of PHBV– line 265-270 – please provide some comment on the one/two melting peaks observed for these materials. What it means? Line 273-274 “The values of IC=O by FTIR and Xc by DSC of the extracted PHBVs gave a poor correlation (R2= 0.53).” – those values are different than those from page 6.

We corrected the value of correlation on page 6 to (R2= 0.53).  We also added more explanations about the reasons for the presence of two melting peaks in the PHBV samples as the following:

Isomorphism phenomena can lead to the presence of two melting peaks [6,31]. As PHBV presents a semi-crystalline structure the crystals with higher HV content will have a higher amorphous phase ratio and thus during heating will be melted (Tm1) first. In crystals with lower HV content, the crystallinity ratio is higher, so the crystals will melt at a higher temperature (Tm2) [31,46]. Other factors that contribute to two melting points include the different crystalline morphologies (thickness, perfection, lamellar stability, or distribution), crystallites of different molar masses, physical aging, and amorphous phase relaxation [31,46].”

  1. Page 9, eq.2 – please provide the explanation for “F” values, is this a fraction value?

Yes. F is the abbreviation of sequence Fraction. We added an explanation of “Fxy” to this section as the following:

FXY represents the mole fraction of XY sequence.

  1. Page 11, table 4 – please correct the table to be more clear.

All the tables are imported from excel and unfortunately, there is no way to make them clearer.

  1. Page 12, Table 5 – please provide the explanation for “K”, “P” and “A, B, X”. You can omit the last column of “sequence distribution” as it is the same for all cases and put this information in the text.

Thank you for the comment. We have explained the parameters of “K” and “P” in the table description. A, B, and X are terms used in the formula of the “mixture of two Bernoullian random copolymers model (model iii)” which are explained comprehensively in the supplementary material. We believe the current format of column of “sequence distribution” is good because the reader can have an easy access to the result of calculation in the table 5 rather than looking for it in the text.

  1. Page 12, table 6 – please provide explanation for “F” values.

Thank you for reminding that. We added an explanation to the footnote of the mentioned table (names table 7 after edit) as the following:

aexptl represent experimental data; (i), (ii), and (iii) are calculated values by Bernoullian model, first-order Markovian model, and mixture of two Bernoullian random copolymers model, respectively.

b3HV molar fraction (mol mol−1) determined by 1H-NMR spectroscopy.

cFX, FXY, and FXVY represent mole fractions of sequence X, XY, and XVY, respectively, where X, Y = V or B.”

  1. Page 15, section 3.7. Mechanical properties – it seems that pilot plant PHBV samples showed lower values of mechanical parameters, as compared to the standard. Please correct the section, especially lines 444-456.

The lower mechanical properties values for the pilot plant PHBV samples compared to the standard sample are because of the higher HV content of the pilot plant PHBV samples which have made it comparable with LDPE. In lines 465-476 the result of mechanical properties of other studies for comparison purposes is also mentioned here.

  1. Page 15, Conclusions, line 467 “…and improved mechanical properties as compared to commercial PHBV (control)..” – they are lower, not improved, please correct.

PHB has a high mechanical property and narrow processing window which makes it brittle and unfavorable for packaging applications. One way of reducing PHB's brittleness is copolymerizing it with HV monomer. It is expected that the PHBV with higher HV content shows lower mechanical properties and a broader processing window compared to PHB which makes it more suitable for flexible packaging. Here, the HV content of the pilot plant PHBV samples was high enough to make their mechanical properties comparable with low-density polyethylene (LDPE) which is favorable for packaging applications. We have revised the text as follows:

“The purified PHBV showed a higher range of thermal properties, broader processing window, and more flexible mechanical properties as compared to commercial PHBV (control), which make the produced PHBV suitable for processing applications such as blow and injection molding”.

Reviewer 2 Report

Generally, the work is on producing PHBV from fermented dairy manure, However, the work is nothing new or interesting. Using waste and mixed cultures for production requires analysis of nutritional components of the carbon sources as it may differ each time and will contribute to the production of 3HV monomers. But there aren't any analysis directed to this. The rest of the analysis are very basic and nothing interesting to be highlighted as well.

Author Response

Generally, the work is on producing PHBV from fermented dairy manure, However, the work is nothing new or interesting. Using waste and mixed cultures for production requires analysis of nutritional components of the carbon sources as it may differ each time and will contribute to the production of 3HV monomers. But there aren't any analysis directed to this. The rest of the analysis are very basic and nothing interesting to be highlighted as well.

Thank you for your comment. Analysis of nutritional components, microbe genomes, and other parameters effective in the production of PHBV and change in HV monomer content which are known as upstream processing parameters are given in our formerly published paper (here as ref 6). In this current manuscript, we covered downstream processing and a comprehensive investigation of the properties of extracted PHBV to see if PHBVs have consistent properties. Also, as it mentioned in the manuscript other studies have focused on PHBV production parameters and optimizing the extraction and recovery yield on a small scale (≤ 3g) or characterizing the properties of PHBV. There is a gap in comprehensive studies examining PHBV biosynthesis using MMC and various feed compositions, extraction of PHBV, and polymer properties at the pilot scale which we covered this gap in our study.

You can find our upstream processing study in the following paper:

“Guho, N. M., Pokhrel, D., Abbasi, M., McDonald, A. G., Alfaro, M., Brinkman, C. K., & Coats, E. R. (2020). Pilot-scale production of poly-3-hydroxybutyrate-co-3-hydroxyvalerate from fermented dairy manure: Process performance, polymer characterization, and scale-up implications. Bioresource Technology Reports, 12, 100588.”

Reviewer 3 Report

1-     Reduce the number of keywords.

2-     There must not be any references in the last paragraph of the introduction, to the aims of the research. Are the authors following other studies' purposes? In that case, there is no novelty.

3-     The introduction can be improved by providing a more critical discussion of recent related literature. Discuss the shortcomings of previous work, the gaps, and how this work intends to fill those gaps. For example, some papers (Renewable and Sustainable Energy Reviews 72 (2017): 95-104; International Journal 4(3) 2013.

4-     Materials and methods: line 85: “as previously reported [6]” must be removed.

5-     In the current state, there are some typographical errors. Therefore, the authors are advised to recheck the whole manuscript to improve the language and structure carefully

6- As a suggestion, a future prospect can also be added at the end of the manuscript.

Author Response

  • Reduce the number of keywords.

We reduced the number of keywords and removed the keywords of “Polyhydroxyalkanoates” and “Downstream processing”.

  • There must not be any references in the last paragraph of the introduction, to the aims of the research. Are the authors following other studies' purposes? In that case, there is no novelty.

There is a gap in study of PHBV studies. Most of studies cover either upstream or downstream parameters of PHBV production. There is a gap in the research stories to cover both groups of parameters and also study of extraction of PHBV in large scale (≥ 3 g). In our research we tried to cover both groups of parameters. In our formerly published paper, we investigated the production parameters (reference 6). In this study, we studied extraction of PHBV in large scale (≥ 3 g) and investigated properties of extracted PHBVs comprehensively. Here is our published paper which covers the production parameters:

Guho, N. M., Pokhrel, D., Abbasi, M., McDonald, A. G., Alfaro, M., Brinkman, C. K., & Coats, E. R. (2020). Pilot-scale production of poly-3-hydroxybutyrate-co-3-hydroxyvalerate from fermented dairy manure: Process performance, polymer characterization, and scale-up implications. Bioresource Technology Reports, 12, 100588.”

  • The introduction can be improved by providing a more critical discussion of recent related literature. Discuss the shortcomings of previous work, the gaps, and how this work intends to fill those gaps. For example, some papers (Renewable and Sustainable Energy Reviews 72 (2017): 95-104; International Journal 4(3) 2013.

Thank you for the comment. We used and added this reference to the draft as the following:

“Depends on the type of renewable waste carbon used as substrate, the type of bacteria and the applied production parameters the production expenses can be reduced by greater than 50% [3,12-15].”

  • Materials and methods: line 85: “as previously reported [6]” must be removed.

Thank you for mentioning that. We removed the “as previously reported” from line 85 but we kept the reference in case a reader be interested in more detailed steps and parameters of production of PHBV can find them in the referenced paper.

  • In the current state, there are some typographical errors. Therefore, the authors are advised to recheck the whole manuscript to improve the language and structure carefully

Thank you for your comment. We have gone through the manuscript and improved the language and structure accordingly.

  • As a suggestion, a future prospect can also be added at the end of the manuscript.

Thank you for the comment. We have mentioned a brief future prospect in the conclusion as the following:

Generally, the purified PHBV showed some variations in properties over the 3 months of operation, which may be attributable to as-yet-refined operational strategies and process controls. To reduce variations at scale would require the operations to be carefully monitored and process controlled in future studies”.

Round 2

Reviewer 1 Report

The manuscript has been corrected,however not each explanation can be accepted. Regarding DSC results, the discussion doesn't include cold crystallization effect, even if it is observed during 1st heating scan. The manuscript worth to be published in the Journal.

Reviewer 2 Report

The paper has to be edited extensively for English